# Fungal Vaccine Development: State of the Art and Perspectives Using Immunoinformatics

**DOI:** 10.3390/jof9060633

**Published:** 2023-05-31

**Authors:** Moisés Morais Inácio, André Luís Elias Moreira, Vanessa Rafaela Milhomem Cruz-Leite, Karine Mattos, Lana O’Hara Souza Silva, James Venturini, Orville Hernandez Ruiz, Fátima Ribeiro-Dias, Simone Schneider Weber, Célia Maria de Almeida Soares, Clayton Luiz Borges

**Affiliations:** 1Laboratory of Molecular Biology, Institute of Biological Sciences, Federal University of Goiás, Goiânia 74605-170, Brazil; 2Estácio de Goiás University Center, Goiânia 74063-010, Brazil; 3Faculty of Medicine, Federal University of Mato Grosso do Sul, Campo Grande 79070-900, Brazil; 4MICROBA Research Group—Cellular and Molecular Biology Unit—CIB, School of Microbiology, University of Antioquia, Medellín 050010, Colombia; 5Laboratório de Imunidade Natural (LIN), Instituto de Patologia Tropical e Saúde Pública, Federal University of Goiás, Goiânia 74001-970, Brazil; 6Bioscience Laboratory, Faculty of Pharmaceutical Sciences, Food and Nutrition, Federal University of Mato Grosso do Sul, Campo Grande 79070-900, Brazil

**Keywords:** bioinformatic, vaccine, fungi, systemic mycosis, neglected disease

## Abstract

Fungal infections represent a serious global health problem, causing damage to health and the economy on the scale of millions. Although vaccines are the most effective therapeutic approach used to combat infectious agents, at the moment, no fungal vaccine has been approved for use in humans. However, the scientific community has been working hard to overcome this challenge. In this sense, we aim to describe here an update on the development of fungal vaccines and the progress of methodological and experimental immunotherapies against fungal infections. In addition, advances in immunoinformatic tools are described as an important aid by which to overcome the difficulty of achieving success in fungal vaccine development. In silico approaches are great options for the most important and difficult questions regarding the attainment of an efficient fungal vaccine. Here, we suggest how bioinformatic tools could contribute, considering the main challenges, to an effective fungal vaccine.

## 1. Introduction

Fungal infections represent a worldwide health problem, with an annual death rate of around 1.5 million individuals, exceeding the number of deaths caused by malaria and being close to that of tuberculosis and the acquired immunodeficiency virus [1]. Furthermore, these agents are responsible for economic losses on the scale of billions of dollars. In 2017, in the United States of America, more than USD 7.2 billion was spent because of the occurrence of these infections [2].

The most worrying global scenario is triggered by systemic mycoses caused by fungi such as *Paracoccidioides* spp., *Histoplasma capsulatum*, *Aspergillus* spp. *Coccidioides* spp., *Candida* spp. and *Cryptococcus*, which are responsible for paracoccidioidomycosis (PCM), histoplasmosis, aspergillosis, coccidioidomycosis, candidiasis and cryptococcosis, respectively. These mycoses in general present as pulmonary disease, skin involvement and disseminate into tissues and systems. Furthermore, invasive fungal infectious diseases have led to concern, especially during the coronavirus disease (COVID-19) pandemic [3]. The emergent mucormycosis disease, caused by fungal members of the *Mucorales* order, causes severe and potentially life-threating fungal infections in immunocompromised individuals, and is a worldwide-distributed species commonly involved in the human diseases *Cunninghamella*, *Lichtheimia*, *Mucor*, *Rhizomucor*, and *Rhizopus* [4].

The knowledge of the fungal biology is crucial for the development of strategies against parasites, which include therapies and vaccines, as well as rapid diagnostic tests. In addition, the antifungal drug arsenal is often limited by toxicity, resistance, and a high cost. To circumvent these difficulties, alternative approaches for their prevention and treatment are being developed, including vaccines and passive immunotherapy [5].

Currently, there are no fungal vaccines approved for use in human beings, but several promising strategies have been developed. Here, we discuss the main methods applied in the development of vaccines and passive immunotherapy against pathogenic fungus. Additionally, we consider the promising application of computer tools based on the immune response in the context of vaccine development, called immunoinformatics, and the future perspectives of its application in fungal vaccine development.

## 2. Vaccine Approaches to Protect against Fungal Infections

### 2.1. Inactivated and Live-Attenuated Vaccines

The inactivated vaccine was the first approach applied in the development of vaccines and involves the inactivation or killing of etiological agents using chemicals, heat, or radiation. A good example of a fungal inactivated vaccine is the use of radiation to produce a vaccine against *P. brasiliensis,* which was found to be able to induce protection and reduce the clinical symptoms and fungemia of mice [6]. In this context, a new method using a heat-killed *Saccharomyces cerevisiae* (HKY) vaccine that can induce protection against non-specific fungal infection is considered to be a special example of a pan-fungal vaccine [7]. The HKY vaccine is effective in protecting CD-1 and BALB/c mice against systemic mycosis caused by *Coccidioides* [8], *C. albicans* [7], *A. fumigatus* [7] and can be used as a therapeutic vaccine [9]. Presumably, *S. cerevisiae* can induce protection against a variety of fungi because it shares with them common polysaccharide epitopes that are present in its cell wall (Table 1).

The first vaccine against coccidioidomycosis, the formalin-killed *Coccidioides immitis* spherules (FKS) vaccine, is another important example of a fungal vaccine in this category that has led to good results in experimental studies [10]. In particular, in a formulation used for intramuscular administration in CD-1 mice, the FKS vaccine demonstrated full protection against *C. immitis* lethal challenge [11]. However, when tested in phase 3 clinical trials, it failed to significantly reduce the incidence of disease or its severity [12]. The HKY vaccine was compared to the FKS vaccine regarding its ability to induce protection against *C. immitis* challenges, inducing 70% and 100% protection in CD1 mice, respectively [8]. These results are similar to those found with a live-attenuated strain of *C. posadasii* (Δcts2/ard1/cts3 or ΔT), which could not endosporulate due to the disruption of two chitinase genes when a triple-attenuated vaccine was employed. This vaccine was able to protect 75–100% of the animals challenged with a virulent C735 strain of *C. posadasii* when using two subcutaneous vaccination injections (14 days interval) [13]. The groups highlighted the fact that the live spores of the attenuated strain showed a low degree of reactogenicity compared to the results for the FKS-vaccinated mice [13]. It is important to note that the *Coccidioides posadasii* CPS1 deletion mutant, which is an avirulent strain, was able to protect over 95% survival, with mean residual lung fungal burdens of <1000 CFU in contrast to an otherwise lethal *C. posadasii* intranasal infection [14]. In the context of a vaccine against coccidioidomycosis, a recent study deserves attention regarding its pioneering work. Mendel et al. (2022) deleted a conserved transcription factor in *Coccidioides*, Ryp1, which plays a dual role in both hyphal and spherule development. Although the vaccination of C57BL/6 mice with live Δryp1 spores was not found to provide any protection from lethal *C. posadasii* intranasal infection, this work identifies the first transcription factor that drives mature spherulation and virulence in Coccidioides [15].

**Table 1 jof-09-00633-t001:** Vaccine-based approaches proposed to protect against fungal infections (Nd; non-determined).

Target Pathogen	Antigen/Strain	Adjuvant/Carrier/Vehicle	Vaccine Type	Model	Route of Injection	Human Clinical Trial	Reference(s)
Paracoccidioidomycosis (PCM)	*P. brasiliensis*	Nd	Inactivated/Live attenuated	Mice	-	Nd	[6]
	Major 43-kDa antigenic glycoprotein (gp43), (P10)	Plasmid vector	DNA Vaccine	Mice	Intramuscular/Intradermal	-	[16]
	*Mycobacterium leprae* derived HSP65	Vector pVAX1/	Recombinant DNA	Mice	Intramuscular	-	[17]
	Major 43-kDa antigenic glycoprotein (gp43), (P10)	Plasmid vector/IL-12 recombinant	DNA Vaccine	Mice	Intratracheal	-	[18]
	Major 43-kDa antigenic glycoprotein (gp43), (P10)	*S. cerevisiae* expressing gp43	Recombinant protein	Mice	Intraperitoneal	-	[19]
	P10- FliC fusion protein	Freund adjuvant (CFA)/multiple-antigen peptide (MAP)	Recombinant protein	Mice	Intranasal	-	[20]
	Recombinant rPb27	Corynebacterium parvum/aluminum	Recombinant protein	Mice	Subcutaneous	-	[21]
	Heat shock protein 60 (HSP60)	Monophosphoryl lipid A, synthetic trehalose dicorynomycolate, and cell wall skeleton	Recombinant protein	Mice	Subcutaneous	-	[22]
Panfungal	β-glucans of *S. cerevisiae*	Nd	Heat Killed Yeast (HKY)	Mice	Subcutaneous	Nd	[23]
Coccidioidomycosis	Formalin Killed Spherules (FKS)	Nd	Whole organism/Inactivated	Human	Intramuscular	Phase 3	[12]
	Antigen 2 (Ag2)	Nd	DNA vaccine	Mice	Intraperitoneal	-	[24]
	*Coccidioides posadasii* CPS1 Deletion Mutant	rAg2/PRA1–106-CSA with MPL-SE (25 μg)/CpG (10 μg) adjuvant	Whole organism/Live-attenuated	Mice	Subcutaneous/Intraperitoneal	-	[14]
	Δcts2/ard1/cts3 or ΔT—triple attenuated vaccine	Nd	Whole organism/Live-attenuated	Mice	Subcutaneous	-	[13]
	Recombinant *Coccidioides* polypeptide antigen (rCpa1) encapsulated into glucan-chitin particles (GCP-rCpa1)	Mouse serum albumin (MSA) and incomplete Freund’s adjuvant	Recombinant protein	Mice	Subcutaneous	-	[25]
Blastomycosis	Adhesin BAD1 gene	Nd	Whole organism/Live-attenuated	Mice (T CD4^+^ depleted)	Subcutaneous	-	[26]
Cryptococcosis	*C. neoformans* strain H99γ	Nd	Live-attenuated	T-cell depleted mice	Nasal inhalation	-	[27]
	Glucuronoxylomannan (GXM)	Tetanus toxoid (GXM-TT)	Conjugate/Solubleantigenic fractions	Mice	Subcutaneous	-	[28]
	*C. neoformans* Δsgl1	Nd	Whole organism/Live-attenuated	Mice	Intranasal	-	[29]
	*C. neoformans* deletion of ZNF2	Nd	Whole organism/Live-attenuated	Mice	Intranasal	-	[30]
Candidiasis	Agglutinin-like sequence 3 (Als3p)	Aluminium hydroxide (Alum)	Recombinant protein (NDV-3)	Mice/Human	Oropharyngeal, Vaginal and Intravenous	Phase I	[31,32]
	Recombinant secretory aspartyl proteinase (r-SAP-2)	Cholera toxin (CT)	Recombinant	Rat	Intravaginal	-	[33]
	PEV7 (r-Sap2 virosomes	Cholera toxin (CT)/Virosomal carrier	Recombinant protein	Mice/Human	Intravaginal	Phase I (delivered by intramuscula)	[34]
	Laminarin (Lam) β-glucan	Complete Freund’s adjuvant (CFA)	Lam- diphtheria toxoid CRM197 conjugate	Mice	Priming dose: Subcutaneous Booster: Intranasal	-	[35,36]
	Fructose bisphosphate aldolase (Fba) (cytosolic and cell wall peptides)	Alum or monophosphoryl lipid A (MPL)	Subunit	Mice	Subcutaneous	-	[36]
	The β-mannose trisaccharide, the Fba peptide T-cell epitope, a dectin-1 ligand, β1,3 glucan hexasaccharide	Freund’s incomplete adjuvant/with and without alum	Conjugate	Mice	Subcutaneous		[37]
	*C. albicans* serotypes a and b ribosomes	Nonencapsulated Klebsiella pneumoniae proteoglycan	Recombinant/Conjugate capsule	Women with vulvovaginal candidiasis (VVC)	Oral	phase II	[38]
Histoplasmosis	Heat Shock Protein 60 (HSP-60)	Monophosphoryl lipid A, synthetic trehalose dicorynomycolate, and cell wall skeleton	Recombinant protein	Mice	Subcutaneous		[39]
	HIS-62	Complete Freund’s adjuvant (CFA) or incomplete Freund’s adjuvant (IFA)	Recombinant protein	Mice	Subcutaneous	-	[40]
	80-kilodalton antigen	Complete Freund’s adjuvant (CFA) or incomplete Freund’s adjuvant (IFA)	Recombinant protein	Mice	Subcutaneous	-	[41]
	H Antigen	Monophosphoryl lipid A, synthetic trehalose dicorynomycolate, and cell wall skeleton	Recombinant protein	Mice	Subcutaneous	-	[42]
Pneumocystosis	Kexin genes	Vector: CMV to express Antigen EF-1α to express CD40L	Kexin-CD40 L DNA vaccine	CD4-deficient mice	Intramuscular	-	[43]
Aspergillosis	*Aspergillus fumigatus* ΔsglA	Nd	Whole organism/Live-attenuated	Mice	Intranasal	-	[44]
	Antigen Asp f 3 and Asp f 9 (VesiVax^®^ Af3/9)	Lipidated Tucaresol, monophosphoryl lipid A or Pam3CAG	Recombinant protein and VesiVax liposomes	Mice	Subcutaneous and inguinal region	-	[45]
Sporotrichosis	ZR8 peptide is from the GP70 protein	Freund’s incomplete adjuvant	Recombinant protein	Mice	Intramuscular	-	[46]

Indeed, the use of live-attenuated vaccine is an approach commonly applied because of their similarity to the effects of the infectious agent, producing a long-term and strong immune response. Currently, there are several studies in this field evaluating live and inactivated fungi with regard to the use of different strategies. However, although live-attenuated vaccines generally have a good safety profile in immunocompetent individuals, they may still cause an infection or a dysregulated inflammatory response in immunosuppressed individuals [47], who are the most susceptible to fungal infections [48]. As a consequence of this, some other studies have been developed in order to improve the immunization of this group. For example, the deletion of the *Blastomyces* adhesion 1 (BAD-1) gene is an attenuated vaccine candidate that is able to induce immunity against blastomycosis in immunocompromised patients [26]. This optimistic result can be explained by the fact that CD8^+^ T cells compensate in the absence of CD4^+^ cells [49] and, alone, mediate efficient antifungal vaccine immunity. Immunity via CD8^+^ T cells was restricted by MHC class I and mediated by the production of cytokines, such as tumor necrosis factor (TNF), interferon gamma (IFN-γ), and granulocyte/macrophage colony-stimulating factor (GM-CSF). This study indicates that CD8^+^ T cells could be a target for robust vaccine-induced immunity against experimental fungal pulmonary infections caused by *Blastomyces dermatitidis* and *H. capsulatum* [26].

Another special live-attenuated vaccine strategy that induced protection against Cryptococcosis used a murine gamma interferon-producing *Cryptococcus neoformans* strain. This is the first instance that a pathogenic fungus has been genetically altered to express a cytokine with biological effects in vivo and that has previously been shown to be protective towards the resolution of disease. The “immune-deficient” A/Jcr mice infected with the IFN-γ-expressing *C. neoformans* H99 not only recovered from the primary infection, but were also completely protected against a second challenge with a pathogenic *C. neoformans* strain, as a result of the protection mediated by IFN-γ-producing CD4^+^ Th1 cells [27]. Recently, immunization with cryptococcal cells overexpressing ZNF2 (a transcription factor related to *C. neoformans* filamentation), either in a live or heat-inactivated form, was found to offer significant protection to the host from a subsequent challenge by the otherwise lethal wild-type H99 strain [30]. In another study, a heat-killed *Cryptococcus neoformans* Δsgl1 mutant accumulating sterylglucosides protected mice in the absence of CD4^+^ T cells, a condition that is most often associated with cryptococcosis. In addition, this vaccine was able to decrease the lung fungal burden and had a robust therapeutic effect [29]. A similar study was conducted against *A. fumigatus.* Vaccination using the strain with the deletion of the sterylglucosidase-encoding gene (*Aspergillus fumigatus* ΔsglA) fully protected immunocompromised mice against a lethal wild-type *A. fumigatus* challenge [44].

It is important to highlight that this is a live vaccine and that there are, therefore, concerns about its use in human individuals. In particular, the commercialization and the safety of attenuated vaccines in immunosuppressed hosts cannot be guaranteed. However, several strategies could be considered for the immunization of CD4^+^ T cell-deficient subjects, particularly HIV/AIDS patients. Furthermore, in immunocompetent individuals, attenuated vaccines against viral infections have been highly successful and perhaps an attenuated vaccine against an endemic fungal pathogen may contribute to the eradication of these diseases where they are prevalent [48].

### 2.2. Recombinant (Subunit) Vaccines

Subunit vaccines are one of the most investigated types of fungal vaccine and consist of one or more purified recombinant protein (or epitope) or polysaccharide of fungi. This approach is very safe when compared to attenuated vaccines. They are especially important when thinking about a vaccine against fungal infection because of the high susceptibility of immunocompromised patients. Because of this, there are a large number of antigenicity studies on various strains, including *P. brasiliensis* [19,21,50], *H. capsulatum* [39,40,41,50], *A. fumigatus* [51,52,53], *C. neoformans* [54,55], *C. immitis* [56,57,58], *C. posadasii* [25,57,58,59,60] and *Candida albicans* [31,33,34,61,62,63]. The scientific basis of this technology comprises the identification of the immunogenic targets, which are obtained via the use of recombinant DNA technology. This involves the insertion of DNA that encodes an antigen (such as a bacterial surface protein) expressed in bacterial, fungal or mammalian cells, purifying it from them, and then applying it in order to trigger the desired immune response. In fact, in this approach, a gene that is transmitted encodes a molecule portion related to the virulence and pathogenicity of a microorganism [64]. It is important to enhance the role of immunoinformatics in this process. This new science comprises a set of computer tools and a database based on immune responses to help in the identification of immunogenic molecules; this makes the vaccine workflow more rational, and reducing the cost and time [65].

When identified, these epitopes or antigens are often combined with an appropriate adjuvant or protein carrier, mostly bacterial toxoids, to establish an efficient immune response and prolonged immunity. For example, NDV-3, an anti-*Candida* vaccine that includes the invasin protein agglutinin-like sequence 3 (Als3p) and alum adjuvant in its formulation [31], prevents yeast attachment and the invasion of epithelial and endothelial cells via IFN-γ and IL-17A-production. This immune response was observed as a result of the improved outcomes in mouse models of *S. aureus* and *C. albicans* infection by inducing upstream Th1, Th17, and Th1/17 lymphocytes, which enhanced the recruitment and activation of neutrophils in infected tissues, thereby reducing the tissue infectious burden [66]. These data are in consonance with the first-in-human Phase I clinical trial [31]. In addition, because of the high homology between Als3p and clumping factor A on the surface of *Staphylococcus aureus*, NDV-3 has also been shown to be protective against this highly virulent bacterial pathogen in animal models [67]. Furthermore, NDV-3 was protective against recurrent vulvovaginal candidiasis (RVVC) [68], and vaccination inhibited the dissemination of *C. albicans* to the kidneys, preventing the colonization of central venous catheters in a murine model of infection [69].

Another promising strategy with regard to a recombinant vaccine against *Candida* is the use of recombinant secretory aspartyl proteinase (r-SAP-2). The administration of a intravaginal vaccine was able to induce the immunoglobulins G (IgG) and A (IgA), leading to protection against an intravaginal challenge with *C. albicans* [33]. Pevion, a Swiss biotech company, has incorporated r-Sap2 into influenza virosomes. This novel vaccine, PEV7 (r-Sap2 virosomes), conferred protection to rats experimentally challenged with *C. albicans* [34]. The clearance of the fungus from the vagina was accelerated and the resolution of the infection occurred at least one week earlier when compared to the administration of empty virosomes and a challenge with the fungus. This vaccine was evaluated in a phase I clinical trial test and demonstrated favorable safety and the generation of specific and functional B cell memory in 100% of the vaccinated women, encouraging the use of this vaccine as potential therapy [70]. The commercial rights of the r-Sap2 vaccine were acquired by NovaDigm Therapeutics Inc. (Grand Forks, ND, USA), which developed the vaccine containing the Als3 antigen (NDV-3) mentioned above. In addition, NovaDigm has acquired the rights to a hyphally regulated protein 1 (Hyr1) vaccine [71] and to a β-mannan conjugate vaccine [72]. The company intends to produce a multivalent vaccine that can induce an immune response against multiple virulence molecules of *Candida* [73]. It is important to note the successful use of the peptide SLAQVKYTSASSI in the Sap2 vaccine via the application of phage display technology, with its innovative character inducing strong immune responses in the animal model [74]. Another example of a recombinant vaccine against *Candida* was developed: a double-peptide construct used to target epitopes derived from fructose bisphosphate aldolase (Fba) and methionine synthase (Met6), which are expressed on the *C. albicans* cell surface, resulting in a high humoral response [75].

In the context of a multivalent vaccine, we highlighted the multivalent recombinant *Coccidioides* polypeptide antigen (rCpa1) that consists of three previously identified antigens (i.e., Ag2/Pra, Cs-Ag, and Pmp1) and five pathogen-derived peptides. The purified rCPA1 was encapsulated into four types of yeast cell wall particles containing β-glucan, mannan, and chitin (in different proportions), or was mixed with an oligonucleotide containing two methylated dinucleotide CpG motifs, showing a high survival rate [25].

A robust in silico analysis of a global proteome of *Candida* using a concept of reverse vaccinology was recently published. The goal of this study was to find vaccine targets using several steps of computational tools to achieve a list of the best targets to be employed in the development of a new vaccine [76]. The study represents the first proteome-wide immunoinformatic approach used to identify the immunodominant epitopes and design a multivalent subunit vaccine against *C. albicans* [77]. Eight antigenic proteins with known functions in hyphal formation (Als4p, Als3p, Fav2p, Als2p, Eap1p, Hyr1p, Hwp1p, Sap2p) were identified. Immunogenicity testing led to the selection of 18 unique epitopes and conservation analysis also showed that the selected epitopes (in addition to the eight hyphal proteins) were present in other *Candida* proteins presenting sequence homology (Sap1p, Sap3p and Als1p). Obviously, it is necessary to confirm the immunogenicity of these antigens via experimental tests. The goal of this approach was to select the most promising targets in a short analysis time and with a low cost for subsequent experimental tests, improving the development of a vaccine with rational drawing, as has been shown for protein-based vaccines against serotype B meningococcal vaccines [78]. In Section 3, we will discuss some of the tools used for the in silico analysis that was conducted in order to identify targets in fungal pathogens for the development of a vaccine by using bioinformatics.

### 2.3. Conjugate Vaccines

Conjugate vaccines are another important approach employed in fungal vaccine development; they involve the association of a weak with a strong immunogenic antigen, commonly a polysaccharide and protein, respectively. The goal of this strategy is to generate a potent immune response to the weak antigen as a result of the B and T cell interaction. The epitopes on polysaccharides can be recognized by B cell receptors and peptides can be recognized by T cell receptors during the process of antigen presentation by class II MHC molecules expressed on B cells, which act as antigen-presenting cells (APCs) to T cells [79].

The first conjugate vaccine developed for fungal infections was against *C. neoformans*, with a capsular polysaccharide, glucuronoxylomannan (GXM), covalently linked to tetanus toxoid (TT) and monophosphoryl lipid A (MPL) as an adjuvant. This vaccine elicited high levels of IgG and IgA specific to GXM and protected 70% of mice after being intravenously challenged. Additionally, passive immunization with antisera from immunized animals also protected naive mice from a lethal inoculum of intravenously administered *C. neoformans* [28]. This study was a pioneer and basis for the development of various forms of a conjugate vaccine against cryptococcosis [80]. Recently, a GXM oligosaccharide structure (a serotype A decasaccharide) was identified for the first time, offering insight into the binding epitopes of a range of protective monoclonal antibodies and furthering our efforts to develop semi-synthetic conjugate vaccine candidates against *C. neoformans* [81].

One major advantage of the conjugate vaccine strategy is that these vaccines are based on targeting the polysaccharide epitopes, which are common in all fungi, especially β-glucans [64]. It is possible to highlight some studies evaluating these molecules, most of them targeting *Candida.* For example, the parenteral administered β-glucan-conjugate vaccine formulated with the human-compatible MF59 adjuvant was assessed in a murine model of vaginal candidiasis. It conferred significant protection, and this was associated with the production of serum and vaginal anti-β-glucan IgG antibodies [82]. Additionally, a glycoconjugate vaccine comprising laminarin (a β-glucan polysaccharide) conjugated with the diphtheria toxoid CRM197 showed protection against both lethal systemic infection in mice, as well as against a self-healing vaginal *C. albicans* infection in rats, showing additional protection against *A. fumigatus* [82]. Similar results were observed with another vaccine, β-glucan-CRM197, formulated with the human-acceptable adjuvant MF59, which conferred protection to mice lethally challenged with *C. albicans* [35]. More recently, the 1,2-linked β-mannose trisaccharide was used in a synthetic conjugate vaccine mixing the Fba peptide T-cell epitope, a dectin-1 ligand, and β1,3 glucan hexasaccharide, stimulating the immune response of mice to a fully synthetic conjugate prepared using these components [37]. Additionally, the same group developed mimotopes that structurally mimic the protective glycan epitope β-(Man)3 as an alternative solution to the complexity of oligosaccharide synthesis [83]. β-glucan, also used in a conjugate vaccine against *A. fumigatus*, was successful in protecting against systemic aspergillosis infection [84].

Other molecules have been used as a conjugate anti-*Candida* vaccine [38,82,85,86]. A good example is the synthetic glycopeptide vaccine β-(Man)3-Fba, constructed by conjugating β-1,2-mannotriose to a peptide segment from fructose-bisphosphate aldolase (Fba), which is a surface antigen of *Candida* spp. This formulation was modified by coupling it to tetanus toxoid (TT), β-(Man)3-Fba-TT, in order to improve immunogenicity and enable it to be used as an adjuvant that is suitable for human use. This modification was crucial for the success of the vaccine, inducing a robust antibody response without the need of an additional adjuvant [36].

In fact, the strategies used to conjugate proteins to polysaccharides enable the immune system to recognize abundant fungal cell wall glycan components, increasing the probability of antibodies recognizing pathogens. Additionally, this strategy can be used to target saccharide epitopes that are conserved in fungal species, particularly β-glucans, thereby creating one vaccine that is effective against a broad range of pathogenic fungi, as a pan-fungal vaccine [48].

### 2.4. Pan-Fungal Vaccine Strategy

Traditional vaccine preparations involve the specificity of adaptive immunity to target one or more antigens found in a single microorganism in order to confer protection. However, current studies support the concept of generating “universal” vaccines that target multiple heterologous pathogens via the use of conserved target antigens [87]. Thus, among the works already cited, there are three vaccines that are able to induce protection against different fungal pathogens: BAD-1 (*B. dermatitidis* and *H. capsulatum*) [26], HKY (*C. posadasii* [8], *C. albicans* [7], *A. fumigatus* [7]) and a conjugate vaccine comprising a β-glucan polysaccharide laminarin conjugated with the diphtheria toxoid CRM197 (*C. albicans* and *Aspergillus* [88]). Moreover, there are robust studies, such as those by Wüthrich and Klein (2011), that have generated T cell receptor transgenic mice (TCR-Tg), also called 1807 mice, using a clone of CD4^+^ T cells isolated from mice infected with *B. dermatitidis*, which can elicit an immune response against *B. dermatitidis* and other dimorphic fungi including *H. capsulatum* and *C. posadasii* [89]. Lastly, calnexin, typically an ER protein, localized on the surface of yeast, hyphae, and spores, has shown promising results for the development of a pan-vaccine. The peptide identified in this antigen induces CD4^+^ T cell responses and it is conserved among the endemic, systemic dimorphic fungi, as well as the clinically important *Aspergillus* species, *Fonsecaea pedrosoi*, and even *Pseudogymnoascus destructans*. Moreover, because of the ability of calnexin to induce the clonal expansion of calnexin-specific CD4^+^ T cells during infection, this vaccine may present an immunotherapeutic effect [90].

In the context of a pan-fungal vaccine, it is important to comment on the use of the *Pneumocystis* endoprotease Kexin (KEX1) from *Pneumocystis jirovecii*, which was previously protective against pneumocystis in a model of HIV and *Pneumocystis* coinfection. As a consequence of the KEX1 sequence being highly conserved among pathogenic fungi, the *Aspergillus*-specific KEX1 recombinant homolog was tested in murine models of combination drug-induced immunosuppression that induced decreasing rates of mortality and a lower lung organism burden. Based on the evidence concerning the protective efficacy of the KEX1, the recombinant pan-fungal protein (NXT-2) was used in murine and non-human primate models of invasive aspergillosis, systemic candidiasis, and pneumocystis, resulting in a decreased mortality and morbidity compared to unvaccinated animals. This study supports the concept of a pan-fungal vaccine and was highly optimistic regarding the induction of immune protection to immunocompromised patients [91].

Collectively, these fungal vaccine development studies suggest the use of strategies that confer protection against heterologous pathogens via conserved epitopes, either by structure or linear sequence, which induce convergent mechanisms of immunity. This potential can be increased with the use of computational strategies [79]. In light of this, there is great optimism regarding the conserved molecular structures that are exposed during fungal growth in host tissues and their application as vaccine candidates.

### 2.5. DNA Vaccines

DNA vaccines are based on cassettes from plasmids plus the cDNA that encodes the desired antigen, driven by efficient eukaryotic promoters and the transfer of the gene-containing plasmid to the host. Subsequently, the expressed antigen induces the desired immune response [92].

Plasmid DNA represents an attractive strategy for developing new vaccines against variable type of pathogens. In fungi, the first report of a DNA vaccination was described in 1999 by Jiang et al., in which a plasmid containing the cDNA of antigen 2 of *C. immitis* was used; this lead to superior efficacy regarding the protection of mice when compared to the recombinant Ag2 vaccine [24]. The next year, Pinto et al. (2000), described the first DNA vaccine against paracoccidioidomycosis, in which a mammalian expression vector carried the full gene of the gp43 of *P. brasiliensis* under the control of the human cytomegalovirus (CMV) promoter. The immunization of BALC/c with this vaccine induced protection against the intratracheal challenge with virulent *P. brasiliensis* yeast cells [16]. One decade later, the administration of the vector pVAX1 carrying the *Mycobacterium* heat shock protein 65 (HSP65) DNA gene was able to reduce the pulmonary fungal burden after infection with the *P. brasiliensis* strain 18 [17]. Additionally, in 2012, the group that evaluated the first DNA vaccine against PCM elaborated a new DNA vaccine against *P. brasiliensis.* They used an expression vector carrying the immunodominant peptide P10 from gp43 and administrated it with or without an IL-12-encoding plasmid. The vaccine, given prior to or after infection with the *P. brasiliensis Pb*18 virulent strain, was able to reduce the fungal burden in the lungs of infected mice [18]. Furthermore, similar results regarding mouse immunization with the pcDNA3-P10 depicted a significant reduction in the mouse pulmonary fungal burden after 30, 60 and 120 days of intratracheal infection challenge [93]. These data suggest that this strategy is promising for the prevention and treatment of PCM.

It is possible to mention another study described by Ivey et al. (2003) that used “expression library immunization” (ELI) to identify a *Coccidioides* gene named ELI-Ag1, which has a protective capacity in BALB/c mice against an intraperitoneal challenge with the arthroconidia of this fungus [94]. Additionally, Zheng et al. (2005) identified, on the surface of *Pneumocystis*, a protein named Kexin and have used it to validate DNA vaccination in CD4-depleted mice. Immunization with plasmid-expressing Kexin under the CMV promoter resulted in significant anti-*Pneumocystis* IgG1 and IgG2a titers in CD4-competent mice, whereas titers were significantly lower in CD4-depleted mice [43].

In the context of the DNA vaccine, it is impossible not to comment upon the success of this approach in the development of the vaccine in the fight against COVID-19 [95]. As a consequence of the ethical implications, particularly with regard to safety and health, until then no DNA vaccine had been approved for human use. However, the success of the platform in the context of COVID-19 allowed science to answer several questions about safety and health, especially concerns regarding the possibility of adverse effects to the use of viral vectors, whether related to the activation of oncogenes [96] or to an excessive immune response [97]. The progress of science enables the rapidity, simple development, reproducibility, thermostability and manufacture of this approach with reduced development costs and risks to be affirmed [95].

## 3. Immune Response against Fungal Infections and Approaches Vaccines

The fungi–host relationship depends on the balance between the characteristics of the fungus, such as its virulence and inoculum size, and the host, such as its genetic background (HLA and SNPs), hormonal and nutritional aspects, age, and comorbidities and associated infections [98]. For instance, acute pulmonary histoplasmosis, which mainly affects poultry farmers, construction workers, travelers and cave explorers, shows that the inhalation of a high fungal burden might cause pneumonia in immunocompetent individuals [96]. On the host side, the hormone estrogen is known to protect women of reproductive age from fungi of the genus *Paracoccidioides* [99]. The presence of cavitary lung injury, as a result of previously treated tuberculosis, is an example of tissue alterations that favor the appearance of chronic pulmonary aspergillosis, also known as aspergilloma or fungal ball [100].

The immune response of fungal systemic mycoses is complex, involving several innate and adaptive immune mechanisms that even overlap due to the chronic nature of these fungal infections [101]. However, the main immunological mechanisms of each fungal disease become more evident when the different types of immunosuppression are associated with the development of different clinical manifestations of systemic mycoses according to the type of immunosuppression. For example, HIV/AIDS patients, who are severely immunocompromised, might manifest disseminated histoplasmosis, pneumocystis pneumonia, esophageal candidiasis, cryptococcal meningitis, coccidioidomycosis and paracoccidioidomycosis [101]. These fungi in the morphology of yeast, in their pathogenic form, are mainly eliminated by macrophages activated by the secretion of IFN-y by CD4^+^ helper T lymphocytes. On the other hand, patients with neutropenia or with functional abnormalities in polymorphonuclear (PMN) cells, especially those with hematological malignancies, are more susceptible to the development of invasive aspergillosis, chronic disseminated candidiasis, and candidemia [102]. These filamentous fungi in their pathogenic form are mainly eliminated by undergoing receptor-mediated respiratory burst and degranulation by PMN cells, which are recruited by Th17 cells [103]. In the context of endemic mycoses, which occur mainly in immunocompetent individuals, such as paracoccidioidomycosis, the impairment of cellular immunity is antigen-specific and the intrinsic host factors that predispose an individual to this type of immunosuppression are poorly understood [104]. An overview of the adaptive immune response to pathogenic fungi is demonstrated in Figure 1.

The development of fungal vaccines has been a challenging task due to the complex nature of fungal immunity, including the varied morphological aspects. As shown, for instance, the immune response to *Cryptococcus* yeast differs significantly from the response to *A. fumigatus* spores. In the former, the development of the Th1 immune response is crucial, while the host defense against *A. fumigatus* is contingent upon T helper responses, followed by the action of neutrophils. To surmount these challenges, researchers have directed their efforts towards identifying specific antigens that can elicit specific and more proper protective immune responses against fungal pathogens. Vaccination with a genetically modified yeast strain of *C. neoformans*, known as H99γ, which produces IFN-γ, has been shown to induce protective immunity against cryptococcosis in mice that lack CD4^+^ T cells [105]. When immune-competent mice were inoculated with *C. neoformans* H99γ and subsequently depleted of both CD4^+^ and CD8^+^ T cells before and during challenge with wild-type (WT) *C. neoformans*, they were fully protected, as evidenced by the 100% survival rate and sterilizing immunity [27]. Various fungal components possess unique abilities to activate Th cell responses in murine vaccination models against *A. fumigatus*. For instance, secreted proteins induce Th2 cell activation, membrane proteins induce Th1/Treg responses, glycolipids activate Th17 responses, while polysaccharides primarily stimulate IL-10 production [53]. Therefore, understanding the type of immune response required for each fungal pathogen and identifying specific antigens are critical in the development of effective fungal vaccines.

In addition to identifying specific antigens that elicit the appropriate immune response, it is imperative to ensure that these antigens do not cross-react with other fungi, particularly those present in the human microbiota, in order to avoid any adverse outcomes [106]. There is a theoretical concern that *Candida* vaccines could disrupt the normal microbiota [106], but targeting specific *Candida* antigens in the invasive hyphal form could minimize this potential drawback [107]. Therefore, prospective studies must take care when selecting antigens with cross-reactivity to microorganisms in consideration of the microbiota. Additionally, the monitoring of the microbiota should be included in preclinical and clinical trials.

### 3.1. Vaccines Based on Antibody

Unfortunately, there are currently no therapeutic vaccines licensed against fungal infections for human or veterinary use. A variety of studies have shown the positive effect of antibody administration against fungal infection, alone or in combination with antifungal drugs [108]. One of the most known studies is on Efungumab (Mycograb^®^), a human genetically recombinant antibody that binds to *Candida*’s HSP90, protecting against several *Candida* species and synergizing with antifungal drugs when evaluated in vitro and in preclinical studies [109]. Furthermore, in a multinational phase II clinical trial, Mycograb^®^ combined with lipid-associated amphotericin B (AMB) improved the overall clinical response from 48 to 84%; this was compared to AMB monotherapy in patients with invasive candidiasis [110]. However, due to production difficulties, as well as safety and quality issues, the authorization of marketing was refused [111].

Another antibody with potential therapeutic use against *Candida* is the IgM mAb C7 (mAbC7), which was produced via the immunization of BALB/c mice with a 200 kDa stress mannoprotein present in the *C. albicans* cell wall [112]. This molecule can react with an Als3p peptide epitope [113], *C. albicans* enolase and the nuclear pore complex protein Nup88 [114]. Additionally, the protection of the NDV-3A vaccine against multidrug-resistant *Candida auris* infection is attributed to anti-Als3p antibodies and to CD4^+^ T helper cells that activate the tissue macrophages. Thus, the mAbs to Als3 may be a powerful therapy by which to combat this emerging fungal pathogen [115].

Recently, the special role of antibodies against fungal cell components, such as mAbs against β-(1→3)-D-Glucan, an essential component of the fungal cell wall, was shown. The antibodies were produced and named mAb 5H5 (IgG3 class) and mAb 3G11 (IgG1 class). Both can interact with yeast and filamentous fungi, including species from *Aspergillus*, *Candida, Penicillium* and *S. cerevisiae*. Furthermore, the mAbs could inhibit the germination of *A. fumigatus* conidia and demonstrated synergy with the antifungal fluconazole in the killing of *C. albicans* in vitro. In addition, mAbs 3G11 and 5H5 demonstrated protective activity in in vivo experiments against *A. fumigatus*, suggesting that these β-glucan-specific mAbs could be useful in combinatorial antifungal therapy [116].

The majority of the study regarding antibody therapy is about *Candida* and *Aspergillus*. However, in a recent review about immunotherapy against systemic fungal infections based on mAbs, it was possible to identify several experiments, in which the use of antibodies in fungal infection was promising, such as in *Sporothrix* spp., *P. brasiliensis*, *B. dermatitidis*, *Pneumocystis* spp. and *H. capsulatum* [117]. Nevertheless, the high cost of the production of antibodies for therapy and the neglected nature of fungal infections [118] means that the use of this approach is very limited and there are no licensed antibodies for therapy against fungal infections [117].

### 3.2. Dendritic Cell Vaccination and Immunotherapy

Dendritic cells (DCs) play an important role in connecting the innate and adaptive immune system after activation by fungal pathogen recognition via their pattern recognition receptors (PRR), phagocytosis of fungal particles, antigen processing and presentation to T helper cells, as well as their secretion of cytokines/chemokines [119]. Thus, DC immunotherapy involves the incubation of DCs ex vivo with selected antigens or pathogens, then returning the cells to the host to boost their protection against an infectious agent. The DC vaccination is like DC immunotherapy, differing with regard to the moment of application. The DC vaccination is performed before infection and DC immunotherapy happens after diagnosis [48]. This immunotherapy approach is mainly used in cancer patients [120]. However, several studies have focused on the use of DC immunotherapy against fungal infections as well [111].

In *A. fumigatus*, DCs have a remarkable functional plasticity in response to conidia and hyphae, showing a capacity to generate antifungal immunity in vivo after activation with live fungi or fungal RNA [121]. Similar results were obtained with recombinant *Aspergillus* proteins and CpG oligodeoxynucleotides (ODNs) as adjuvants [122]. In another study, the vaccine formulation of the DCs transduced with an adenovirus vector encoding the cDNA of IL-12 and pulsed with heat-inactivated *A. fumigatus* induced a protective immune response against invasive pulmonary aspergillosis [123]. Additionally, the DC immunization approach proved to be a powerful way of overcoming the relatively weak immune response of the mouse to the defined small carbohydrates and peptide antigens of *C. albicans* [86]. More recently, DCs pulsed with an acapsular *C. gattii* were used in a vaccine against *C. gattii* and were able to induce cytokine-producing CD4^+^ T cells and multinucleated giant cells, which were associated with protection against pulmonary cryptococcosis in an experimental model [124].

Another DC vaccine approach was developed against *P. brasiliensis*. Bone marrow DCs were pulsed with peptide P10, derived from *P. brasiliensis* glycoprotein 43 (gp43), and administrated subcutaneously to naive mice that were subsequently challenged. An increased production of IFN-γ and IL-12 was observed, along with decreased pulmonary damage and significantly reduced fungal burdens, suggesting that this strategy can be therapeutic as well as prophylactic [125].

### 3.3. Vaccines Based on T Helper Lymphocytes

The CD4^+^ T cells, more specifically T helper and Th17, play major roles in eliciting protective and inflammatory responses [126]. In fungal infection, these cells generally play roles in both the resolution and worsening of superficial or invasive infections [127]. In addition, CD8^+^ T cells have an important role in the immune response against fungal infection [26]. Cell-based vaccines designed to prevent invasive fungal infections are currently being investigated in clinical trials and their use could play an especially important role in AIDS patients [128].

The goal of T cell vaccines or immunotherapy is to induce CD4^+^ and/or CD8^+^ T cells of sufficient magnitude and the necessary phenotype or effector functions that directly contribute to pathogen clearance via cell-mediated effector mechanisms [129]. This approach is used to treat mainly cancer and chronic infections via the intravenous injection of autologous T cells, which have been stimulated in vitro using antigens or modified using a gene encoding a specific antigen receptor, and expanded to a large quantity before being infused back into the patient [130]. For fungi, these strategies have been focused on vaccine studies based on the T cell-mediated immune response [64], predominantly against candidiasis and aspergillosis [64,131].

The treatment of immunocompetent mice with *Aspergillus* crude culture filtrate antigens resulted in the development of local and peripheral protective Th1 memory responses, mediated by antigen-specific IL-2- and IFN-γ-producing CD4^+^ T cells capable of conferring protection upon its adoptive transfer to naive mice [132,133]. The adoptive transfer of Asp f16 peptide-specific CD8^+^ T cells significantly extended the overall survival time of the *A. fumigatus*-infected immunocompromised mice [133]. In BALB/c mice, the cell glucanase Crf1 from *A. fumigatus* was found to induce memory CD4^+^ Th1 cells and cross-protection against lethal infection with *C. albicans* [51].

It is clear that Th cell-mediated vaccine responses or immunotherapy are a promising alternative regarding the treatment of fungal infection, mainly in patients who have undergone a hematopoietic stem cell transplant and chemotherapy. However, more studies are needed to determine the impact of this approach on the host organism and prognosis [134].

## 4. HLA and Its Importance in Identification of Therapeutic Epitopes

Human leukocyte antigens (HLAs) are part of the immunoglobulin gene family. These genes are extremely important in the immune response and are found on the short arm of human chromosome number 6 (region 6.21.3). The presence of these protein molecules in the cell membrane results in the control of immune responses. The presence of these protein molecules in the cell membrane results in the control of immune responses. These molecules operate via cell–cell interaction, acting in the recognition of what is proper and what is not proper for the human system, and can be considered as the hands and eyes of the immune system. HLAs are antigens classified as belonging to class I (HLA-A, -B, -C, -E, -F, -G and -H genes), II (HLA-DR, -DP and -DQ genes) and III (includes genes encoding the complement system and TNF), and are extremely polymorphic and show genetic variability from one population to another [135].

For the selection of the best candidates for vaccines, some characteristics of the immunogenic epitopes must be taken into account. These include stability and hydrophilicity, since these characteristics can influence the synthesis of these molecules. Another important characteristic is the solubility of the antigen, since this characteristic can directly influence the recognition of these vaccine candidate molecules by antigen-presenting cells (APCs) [136]. Finally, verifying the levels of interaction between the epitopes and HLA molecules (class I or II) enables the effective selection of targets for further experimental analysis. Currently, with advances and improvements in immunoinformatics tools, these characteristics mentioned above can be evaluated, as well as predicted [137,138,139,140]. In this review, these prediction tools are cited. Next, some studies involving HLA analyses related to fungal infection processes will be highlighted, as well as their use in the identification of new therapeutic targets. It is noteworthy that all HLA or MHC alleles described in this study are shown in Table 2.

**Table 2 jof-09-00633-t002:** HLA and MHC alleles expressed during fungal infections (+; present/−; absent).

Target Pathogen	HLA	Bioinformatics	Experimental	Model	Reference(s)
*Paracoccidioides* spp.	A1	-	+	Human	[135]
	A2	-	+	Human	
	B7	-	+	Human	
	B21	-	+	Human	
	CW1	-	+	Human	
	B15	-	+	Human	
	A9	-	+	Human	[141,142,143,144]
	B13	-	+	Human	
	B22	-	+	Human	
	B40	-	+	Human	
	B40	-	+	Human	[141,144,145]
	DRB1-0101	-	+	Human	[145]
	DRB1-0301	-	+	Human	
	DRB1-0401	-	+	Human	
	DRB1-0701	-	+	Human	
	DRB1-1101	-	+	Human	
	DRB1-1301	-	+	Human	
	DRB1-0404	-	+	Human	
	DRB1-0802	-	+	Human	
	DRB1-0205	-	+	Human	
	DRB1-1302	-	+	Human	
	DRB1-1501	-	+	Human	
*Histoplasma* spp.	B7	-	+	Human	
	B7	-	+	Human	[146]
	DR-15	-	+	Human	
	DQ-6	-	+	Human	
*Cryptococcus* spp.	DR4	-	+	Mouse	
	C1203	+	-	Human	[55,147]
	DRB1-0101	+	-	Human	
*Coccidioides* spp.	DRB1-0401	-	+	Mouse	[148,149]

### 4.1. Paracoccidioidomycosis

In studies developed by Dias et al. [135], HLA mapping was performed in patients affected by the chronic form of paracoccidioidomycosis (PCM). Twenty-one male patients were analyzed, showing an increase in the frequency of alleles at locus A (14 alleles), B (19 alleles) and C (10 alleles) of the HLAs. Among the identified HLA alleles, some showed significant increases in their frequency and were considered strongly linked to PCM, such as HLA-A1, -A2, -B7, -B21 and -CW1. HLA-B15 also appeared with increased frequency in the patients affected. However, HLA-A1 alleles in PCM are related to an immunological deficiency, which could be related to an inhibitory response or even to the success of an infection caused by *Paracoccidioides* spp. [141]. However, other studies have shown that HLA-A1 is involved in the phagocytosis processes of the fungus caused by neutrophils [150]. Other studies have shown the positive association between *Paracoccidioides* sp. and the expression of HLA-A9, B13, B22 and B40 [141,143,144].

In silico prediction analyses of peptides that bind to HLA-DR molecules (Class II) enabled the identification of the immunodominant epitopes of the 43-kDa glycoprotein (GP43) [151]. For the prediction of immunodominant T cell epitopes in GP43, the TEPITOPE algorithm was employed, using nine different HLA-DR alleles for analyses. With this, five of the most promising epitopes were predicted and selected, and then these molecules were tested in proliferation assays using peripheral blood mononuclear cells (PBMC) from patients with PCM after chemotherapy and PBMCs from control patients. In total, 14 out of 19 patients recognized at least one of the promiscuous epitopes predicted using immunoinformatics [151]. This highlights the importance of HLA-based immunogenic epitope prediction algorithms, demonstrating that bioinformatics tools can lead to promising results. In other studies, the immune response to synthetic GP43 epitopes that bind to several HLA alleles was verified. All epitopes were predicted using bioinformatics tools and synthesized. Then, the responses of peripheral blood T lymphocytes from 29 patients with PCM to the synthesized peptides were evaluated [152]. After analyses, it was found that all patients were typed for HLA class II, and a great diversity of HLA-DR molecules were associated with the recognition of the analyzed GP43 epitopes. In studies carried out by Travassos et al. [153], the immunization potential of the P10 peptide of GP43 was analyzed. In immunization assays, using the P10 peptide, it was observed that this molecule provided protection to Balb/C mice against intratracheal infections caused by virulent strains of *P. brasiliensis*. In addition, immunoinformatics analyses were used, making it possible to verify that 21 HLA-DR molecules could recognize and bind to the P10 peptide.

HLA frequency identification and analyses were also extended to studies of PCM in the central nervous system (NPCM), where six patients with NPCM had a HLA class I and II frequency, assessed by means of microlymphocytotoxicity. However, it was observed that the frequency of HLAs found in the studied patients (HLA-A, -B, -C, -DR and -DQ) was similar to alleles found in other populations affected by PCM [154]. However, among the HLAs analyzed, the increase in the frequency of HLA-B40 stood out. In other studies, this molecule has been reported to be involved in the development and progression of the systemic form of PCM [141,144].

In studies carried out by Mamoni et al. [145], the HLA-DR of PCM infection (PI), in its adult forms (AF) and juvenile forms (JF), was evaluated. However, a higher frequency of HLA-DR was observed in PI patients when compared to JF and AF patients. In other studies, the functional characterization of the P27 protein of *Paracoccidioides* sp. was carried out, where bioinformatics tools were used to trace the immunogenic profile of this molecule. For this, HLA class II molecules (DRB1-0101, -0301, -0401, -0701, -1101, -1301, -0404, -0802, -0405, -1302, -1501 and DRB5-0101) were used to probe the entire P27 protein sequence (220 amino acids). Interestingly, four peptides of P27 showed a high recognition affinity for nine of the twelve HLA-DR molecules selected for analysis [155].

### 4.2. Histoplasmosis

In studies carried by Braley et al. [146], eighteen patients who presented hemorrhagic macular lesions or peripapillary lesions originating from histoplasmosis had their HLA frequency identified. A total of 78% of the patients showed an increase in the frequency of HLA-B7. Furthermore, it was shown that the imbalance between the interaction of HLA-B and genes of the D locus is directly related to immune response genes. Locus B HLAs are involved in the specificity of cytotoxic T cells. This demonstrates that HLA-B may be linked to responses mediated by T lymphocytes, thus being more effective against invading organisms [146].

In studies related to presumptive ocular histoplasmosis syndrome (POHS), this disease has been found to be associated with other HLA alleles within the HLA-A, -B, -DQ, and -DR loci. A total of 34 patients diagnosed with POHS had their DNA analyzed for HLA gene typing. Significant associations were observed between sick patients and HLA-B7, HLA-DR15 and HLA-DQ6. Thus, it was concluded that HLA-DR15, -B7 and -DQ6 are strongly associated with the development of POHS, suggesting that these alleles help regarding individuals’ susceptibility to histoplasmosis [156].

In studies carried by Kischkel et al. [156], epitopes of *Histoplasma capsulatum* were investigated, in order to be used in the construction of diagnostic tests, or in the identification of molecules with vaccine potential. Thus, epitopes of *Histoplasma* sp. that would be recognized by human HLA class I and class II molecules were analyzed. After these analyses, they selected promising epitopes from molecules such as HSP60, Enolase and ATP-dependent molecular chaperone HSC82. The initial assays demonstrated the proliferation of CD4^+^ and T CD8^+^ lymphocytes, in addition to inducing the production of cytokines IFN-γ, IL-17 and IL-2, representing a Th1 and Th17 cell profile; this demonstrates the potential of immunization processes to activate a cellular response [157]. HSP70 of *Histoplasma* sp. also had its immune system excitation potential investigated, where, through immunoblotting and using anti-HLA-DR, the interaction between HLA-DR and HSP70 was shown [158].

### 4.3. Cryptococcosis

The development and construction of a vaccine for immunization against cryptococcosis has become a priority, due mainly to an increase in the number of cases registered in individuals with weakened immune systems [55,159,160]. In studies carried out by Specht et al. [55], glucan particles (GPs) were used as a delivery system for the antigens to be studied. Initially, six recombinant antigens were analyzed, but only four (GP-Cda1, GP-Cda2, GP-Cda3 and GP-Sod1) were used in the studies. Mice of the C57BL/6, BALB/c and HLA-DR4 lineage (transgenic) were challenged with virulent strains of *Cryptococcus neoformans* and *Cryptococcus gattii*, and subsequently immunized with the recombinant antigens. After immunization and analysis, it was observed that the recombinant antigens provided greater protection and chances of survival to the animals. For HLA-DR4 mice, the combination of GP-Cda1 and GP-Cda2 induced protective responses, similar to the responses observed in C57BL/6 mice, thus demonstrating the importance of human HLA-DR4 alleles in relation to the immunization process of these animals [55].

In other studies, the prediction of T cell epitopes was carried out with the aim of synthesizing peptide vaccines against *C. neoformans*. Among the candidate epitopes, the peptide YMAADQFCL showed interaction with nine MHC-I alleles and HLA-A*02:01 alleles. On the other hand, the peptides YARLLSLNA, ISYGTAMAV and INQTSYARL were predicted to bind with MHC-II, representing the central high-binding affinity epitopes. The predicted peptide, including YMAADQFCL and ISYGTAMAV, had an average population coverage of 69.75% and 74.39%, thus demonstrating the importance of immunoinformatics analysis. Such data may help in the construction of a polypeptide vaccine aimed at the immunization or therapy of cryptococcosis [147].

In addition, the use of immunoinformatics has also shown epitopes of HSP70 T cells from *C. neoformans*, with the aim of identifying new vaccine candidates [147,161]. Initially, 10 promising epitopes were predicted and verified by using computational tools, all of which triggered both the cellular and humoral immune response. Furthermore, other analyses found that the predicted epitopes had a population coverage of 90%. These data were obtained through molecular docking, using HLA-C*12:03, HLA-DRB1*01:01 and Immunoglobulin G as a model. of the strength of the association between HLA binding sites and promiscuous peptides were also performed, thus highlighting the importance of HLA-based epitope predictions [147].

### 4.4. Coccidioidomycosis

Studies involving epitopes from *Coccidioides* sp. in the process of immunizing mice and humans have already had their effectiveness evaluated as a vaccine for coccidioidomycosis [147,162]. Other studies have shown infection assays by inoculating spores of *Coccidioides posadasii* in HLA-DR4 transgenic mice (DRB1-0401 allele), which expresses human class II HLA, thus inducing the cellular response of T CD4^+^ lymphocytes. The vaccinated transgenic mice exhibited three distinct clinical manifestations: acute fatal disease, disseminated disease and pulmonary disease. In addition, immunized mice showed activation of heterogeneous immunity against pulmonary infection by *Coccidioides* sp., expressing Th1 and Th17 profile cytokines. The recruitment of innate immunity cells was also observed after nine days of the immunization process. The same immunization procedure was performed in C57BL/6 mice, where 100% survival was observed for mice of this strain [148].

In other studies, the use of a recombinant multivalent vaccine (rCpa1) induced protection against *C. posadasii* and *C. immitis* in HLA-DR4 transgenic mice (DRB1*0401) and in C57BL/6 mice. A significant reduction in the fungal burden was observed in animals initially immunized and subsequently challenged with fungal infection. There was also an increase in the production of IFN-γ and IL-17, represented by the Th1 and Th17 cell profiles [149].

## 5. Bioinformatic Tools for Vaccine Development against Fungi

There are seven challenges posed to the development of an effective vaccine against fungal human diseases: (i) the population most at risk is immunocompromised people; (ii) the diverse sites of infection in the host; (iii) intraspecies and interspecies antigenic variation among fungi; (iv) molecular similarities between fungi and animalia kingdoms; (v) translation from animal models to humans; (vi) formulation; and (vii) commercialization [107]. Among these seven points, bioinformatics can act strongly in most of them. These tools have the potential to perform analyses that are not possible when using traditional approaches. By employing specific tools, it is possible to identify the most promising antigen or epitope from different stages of the fungal infection, as well as those specific for each species without homology in humans. In addition, the immunoinformatic tools are able to predict the epitope using the MHC molecules associated with the fungal infections, specifically in humans (HLA, human leukocyte antigens) [163]. Thus, the clusters of tools and biotechnology can generate a more effective and safe vaccine, which is especially important in the context of immunocompromised individuals [164].

Bioinformatics has an important role in the rational identification of targets, especially in the context of pathogens with a complex life cycle. However, it is relevant to point out that the rational answer to the initial questions is strongly associated with the pathogen involved. Thus, in the fungal infections caused by a large number of species with different characteristics, it is extremely difficult to determine the general workflow that can obtain the specificity of all pathogenic fungal species. In addition, the majority of these infections are superficial and easy to treat, but roughly 150 million cases might be serious or life-threatening to individuals [1]. Invasive fungal infections kill over a million people around the world [165]. In this sense, a promising vaccine target needs to have some main characteristics that allow an immune response against the infection to be elicited, such as virulence, an involvement in the invasion of the host, adherence or phenotypic switch, localization at the cell membrane or an ability to be secreted, as well as the capacity to evade the host's immune system [166].

In late 2022, the World Health Organization (WHO) released a list of fungal priority pathogens with the aim of moving research efforts and investment towards fungal infections and antifungal resistance. This list was organized into three groups—critical, high, and medium priority—based on the rate of mortality, incidence over the last 10 years, geographical distribution, availability of diagnostics and treatments, transmissibility, drug resistance, and complications of the disease. On the other hand, the causative agents of *Nakaseomyces glabrata*, *Fusarium* spp., *Candida parapsilosis*, *Histoplasma* spp., *Mucorales*, *Candida tropicalis* and *Eumycetoma* belong to the high-priority group [167]. Some of them are listed in Table 3, which shows the most recent study in fungal vaccine development using bioinformatics.

Unfortunately, there is only one database of immunoinformatics that specifically concerns fungi FungalRV [168]; this is a server that has gathered several tools, including an adhesin predictor, cellular localization predictor, linear and conformational B cell epitope predictor, and T cell epitope predictor. One detailed protocol of FungalRV can be found in the study of Chaudhuri and Ramachandran (2017) [65].

**Table 3 jof-09-00633-t003:** Bioinformatics as a tool for fungal vaccine development.

Pathogen	Subcellular Location and	B Cell Epitope Prediction	T Cell Epitope Prediction	Cytokines	Immunogenicity and Antigenic	Number of Final Targets	Year	Ref.
*Histoplasma capsulatum*	PSORT II; McGeoch method; TMHMM	--	--	--	VaxiJen 2.0,	5 targets	2023	[169]
*Candida auris*	TargetP; SignalP; Phobius; FunsecKB; PredGPI; TMHMM; EffectorP; FungalRV; FaaPred;	--	NetMHCII 2.3 (IEDB)	--	VaxiJen server	39 targets	2022	[170]
*Rhizopus delemar*	SignalP; PredGPI; TMHMM; GPI- anchor	BCPREDS; Ellipro tool	IEDB (MHC class I and II); MHC class I processing	IL-4Pred; IL-10Pred; IFNepitope	VaxiJen 2.0	4 targets	2022	[171]
*Sporothrix brasiliensis*	--	Bepipred 2.0	Pred^BALC/C^ server;	IL-4pred; IFNepitope; 17eScan server;	SsEno	Enolase	2022	[172]
*Cryptococcus neoformans var. grubii*	--	IEDB Bcell epitope prediction tool; BepiPred; ElliPro	IEDB MHC-I prediction tool; IEDB MHC-II prediction tool	--	Kolaskar and Tongaonkar antigenicity method	heat shock 70 kDa protein	2021	[147]
*Candida glabrata*	--	ElliPro; Bepipred tool from IEDB;	IEDB MHC I prediction tool/IEDB MHC II prediction	--	Kolaskar and Tongaonkar antigenicity method	Fructose Bisphosphate Aldolase	2021	[173]
*Candida dubliniensis*	--	--	IEDB B-cell epitope prediction tool; NetMHCII 2.3; NETMHCpan 4.0 web servers	IL2Pred, IL4Pred, and IFNepitope	VaxiJen 2.0; AllergenFP	Secreted aspartyl proteinases (SAP) proteins	2023	[174]
*Candida glabrata*	SignalP-5; DeepLoc-1.0	--		--	VaxiJen v2.0 server	33 targets	2022	[175]
*Aspergillus fumigatus*	--	--	NetMHCIIpan ver.3.2 server;	--	AllergenFP; VaxiJen ver.2.0	5,8-linoleate diol synthase; ChainB-chitinase A1	2022	[175]
*Rhizopus microsporus*	SignalP-5.0 server	--	IEDB MHC I prediction tool/IEDB MHC II prediction; Docking by AutoDock Vina	INF predictionserver	--	Spore coat (CotH) and Serine protease (SP) proteins as	2021	[176]
*Candida albicans*	CELLO2GO	--	NetCTL server; IEDB MHC I prediction tool/IEDB MHC II prediction	--	VaxiJen server, ANTIGENpro; AllerTOP; NetChop3.1; MHCII-NP	Als4p, Als3p, Fav2p, Als2p, Eap1p, Hyr1p, Hwp1p, Sap2p	2020	[77]
*Candida auris*	CELLO	ABCPred; Ellipro service	NetCTL 1.2; IEDB MHC II prediction	IFNepitope	VaxiJen server; Algpred server	Mitochondrial import receptor subunit, Putative beta-glucanase/Beta-glucan synthetases, 1,3-beta-glucanosyltransferase, Uricase, and a putative SUN family protein.	2022	[177]
*Rhizopus delemar*	TMHMM v2.0 server	IEDB Bcell epitope prediction tool (BepiPred and ElliPro)	NetCTL 1.2; IEDB MHC II prediction	IFNepitope; IL4pred; IL10pred	VaxiJen server; AllerTOP v2.0; MHCII-NP (IEDB); NetChop3.1	Cell membrane by the copper oxidase-iron permease (FTR1) complex	2022	[140]
*Candida tropicalis*	CELLO2GO; PSORT II	--	NETMHC 2.3; NETMHC 4.0; Bepipred (IEDB)	IFNepitope	VaxiJen 2.0; AllergenFP version 1.	Secreted aspartic protease 2 (SAP2) protein	2022	[166]

Next, we summarized some aspects of the bioinformatic tools that could be used in the investigation and development of a promising vaccine against fungal infections. Figure 2 proposes the summarized workflow by which to predict targets for a fungal vaccine based on MHC, B and T cell epitopes and an antigenicity analysis.

### 5.1. T Cell Epitope Prediction

In the context of fugal pathogens, the main mechanisms involved in the immune response against infection are phagocytosis and the activation of the adaptive immune response via the development of different CD4^+^ T helper and regulatory T cells [64,127]. In this way, the identification of HLA class II epitopes in fungal pathogens is a great approach to the development of vaccines and immunotherapy that can be used to elicit antigen-specific immune responses [77]. Nevertheless, via the cross-presentation mechanism, APCs can present extracellular antigens through HLA class I molecules as well [178].

In general, the antigen presentation mechanism with HLA class I and II occurs via the proteolytic cleavage of pathogen proteins into small peptides of 8–14 and 15–35 amino acids, and their binding to the peptide-binding cleft of class I and class II MHC receptors, respectively. This is followed by the positioning of this peptide/MHC complex on the cell surface and the subsequent interaction with T cell receptors. In this context, it is important to point out that the HLA class II binding predictions are currently slightly less accurate than the HLA class I binding predictions because they involve conformational criteria. In addition, HLA class II epitopes are longer (around 15 to 25 mer) and several binding registers or cores may be present in the same peptide. A peptide needs to be presented by an MHC I molecule for it to be able to elicit effector T cell responses. Contrarily to MHC II molecules, which can bind to peptides that are longer and more variable, MHC I binding is restricted to peptides typically 8–14 amino acids long in sequence; in addition, some of the residues in the peptide, denoted as anchor residues, are important for peptide–MHC binding [179].

Before commenting upon the tools, it is important to explain the steps involved in the analysis of T cell epitope prediction. Briefly, the identification of epitopes depends on the interaction of them with the binding region in the HLA molecule. Thus, it is important to define the HLA allele for analysis and the methods used for prediction. Firstly, the choice of allele, which defines the vaccine coverage, can be solved using a list of those alleles that represent the reference sets that should provide >97% and >99% in the human population, for class I and II, respectively; these are available in The Immune Epitope Database (IEDB), the biggest database of immunomic and host tools, that can be referred to in order to assist in the prediction and analysis of epitopes [180]. Furthermore, the MHC-II binding predictions from IEDB are available with seven alleles (DRB1*03:01, DRB1*07:01, DRB1*15:01, DRB3*01:01, DRB3*02:02, DRB4*01:01, DRB5*01:01), which have the best results in the definition of the IEDB consensus percentile rank [181]. In the context of systemic mycosis, it is important to point out that several studies have demonstrated the influence of HLA genes on the susceptibility of infections, highlighting the importance of certain alleles in the analysis [182]. The other point is the choice of the prediction method, which is more difficult because currently there are several online tools can be used to predict the T cell epitope [183]. Therefore, here we introduce several tools for the epitope prediction of MHC class I and II molecules.

Firstly, the set of tools used for the prediction of T cells, called T Cell Epitope Prediction Tools, from IEDB, has great potential. This database provides tools for MHC class I and class II epitope prediction, MHC I processing (Proteasome, TAP) and the analysis of immunogenicity to epitope class I binding [184]. Currently, the methods recommended by IEDB for MHC epitope prediction are the consensus approach, combining NN-align, SMM-align, CombLib, and Sturniolo if any corresponding predictor is available for the molecule, otherwise, NetMHCIIpan is used. Also used are the NetMHCpan (4.0) for MHC I and the NetMHCIIpan-4.0 server [185] for MHC II, which are based on artificial neural networks (ANNs). This method can obtain accurate results for molecules using little or no experimental data and provides more than 200 MHC I molecules, employing binding affinity and eluted ligand mass spectrometry for human (HLA-A, B, C, E), mouse (H-2), cattle (BoLA), primate (Patr, Mamu, Gogo) and swine (SLA) molecules [186]. For MHC II epitope prediction, a combination of experimental and in silico data is used, covering the three human MHC class II molecules, HLA-DR, HLA-DQ and HLA-DP, as well as the H-2 mouse molecules. It can perform a prediction for any MHC II molecule of a known sequence.

In the context of fungi, it is possible to highlight a study focusing on *C. albicans* [77] and *Aspergillus flavus* [187]. The tool was used in the global analysis of the *C. albicans* proteome and was able to identify eight antigenic proteins with comparative functions in hyphal formation (Als4p, Als3p, Fav2p, Als2p, Eap1p, Hyr1p, Hwp1p, Sap2p) and select 18 epitopes that are conserved among 22 *C. albicans* strains [77]. In addition, more recently, it was used in a robust immunoinformatic analysis of the HSP70 kDa protein complex of *C. neoformans* var. *grubii*, with a promising epitope identification and a massive global population coverage (based on the allele frequency and geographic distribution) [147]. The immunoproteomic from *Sporothrix brasiliensis*, with subsequent epitope prediction using the IEDB MHC II tool, is a good example of the bioinformatics that can be applied to vaccine development against fungal infection. ZR8 peptide from the GP70 protein, the main antigen of the Sporothrix complex, was the best potential vaccine candidate, inducing a strong cellular immune response [46]. In addition, another important study of mucormycosis was based on the reverse vaccinology approach. Six final proteins were identified from a total of 29.447 proteins obtained by Uniprot. This set of proteins was subjected to adhesin prediction, localization prediction, immunogenicity analysis, analysis of their homology with humans, allergen analysis, and T and B cells analysis via IEDB in order to obtain the final targets [188].

Another immunoproteomic study that used a group of combined immunoproteomic and immunopeptidomic methods, based on co-immunoprecipitation, to map *H. capsulatum* epitopes for the first time in a natural context using murine dendritic cells and macrophages can be highlighted. Additionally, a robust in silico analysis was used to predict MHC I and II epitope binding from human and mice, as well as immunogenicity and IFN-γ induction prediction. The four most promising peptides, derived from heat shock protein 60, enolase, and the ATP-dependent molecular chaperone HSC82, were synthesized, as well as the peptides with and without incorporation into glucan particles that induced a strong immune response in the vaccination. The authors point out the fact that these proteins have a high degree of identity with the proteins expressed by other medically important pathogenic fungi, which is interesting in the context of the pan-fungal vaccine [157].

In addition, it is possible to highlight ProPred1, which is an online tool used for the prediction of the MHC I epitope. It implements matrices for 47 MHC Class I alleles, and proteasomal and immunoproteasomal models [189]. Similarly, RANKPEP identifies epitope binding to MHC I and MHC II molecules from protein sequences or sequence alignments using position-specific scoring matrices (PSSMs) [190]. Likewise, nHLAPred is an MHC class I epitope prediction tool that can identify epitopes binding to 67 alleles and also allows the prediction of proteasome cleavage at the C-terminus [191].

For MHC class II molecules, another important tool is TEPITOPEpan, which is similar to NetMHCIIpan (the current version (NetMHCIIpan-4.0) [186]) and is able to predict epitope binding with very restricted or no experimental data; it has been evaluated as the second-best method after NETMHCIIpan-2.0 (among the four pan-specific methods: NetMHCIIpan-2.0, NetMHCIIpan-1.0, MultiRTA and TEPITOPEpan) for predicting the binding specificities of an unknown allele. In the context of MHC class II, TEPITOPEpan was applied to the gp43 antigen from *P. brasiliensis* in a key study. In another example, the application of the recombinant *C. neoformans* chitin deacetylase 2 (Cda2), that has been discussed before, in a vaccine against fungal infection; it was revealed a peptide sequence predicted to have strong binding to the MHC II, H2-IAd allele was found in BALB/c mice. The protection was lost after the induction of a mutation in the sequence of Cda2, indicating that the immune response is dependent on the strong binding of the Cda2 [192].

Lastly, a recent review of the methods used to predict the epitope binding of MHC suggested that the combination of sequence and structure-based methods would be the best method, but that this is hindered by the lack of 3D structures [191]. However, it is possible to use an approach employing different tools based on a sequence and 3D structure. An example of this is the possibility of combining the sequence-based tools, as mentioned here, with an EpiDock, the first structure-based server for MHC class II binding prediction. This tool can predict binding to the 23 most frequent human alleles and in one study, was able to identify 90% of true binders and 76% of true non-binders, with an overall accuracy of 83% [183].

Therefore, based on the tools discussed here, Figure 2 proposes a protocol that can be used to analyze the prediction of MHC molecule epitopes.

### 5.2. B Cell Epitope Prediction

B cell epitope prediction is essential in the context of the modern analysis and development of vaccines and diagnostics. B cell epitope mapping is essential in the production of diagnostic tests, although it is only the first step in designing potent vaccines. In addition, B cells are able to recognize linear (continuous) and conformational (discontinuous) epitopes. The linear epitopes have their amino acid residue organized in the primary sequence of the protein, while the discontinuous epitopes are formed by residues organized far in the primary structure, but become nearer as a consequence of the folded protein [185]. Linear epitope prediction is more simple than conformational prediction and normally the amino acid sequence is required; however, it represents only 10% of the B cell epitopes. On the other hand, the conformational epitopes represent 90% of the total B cell epitopes but present the difficulty of prediction, especially in cases of neglected tropical diseases; this is because they frequently require the PDB format as input [185], with a few exceptions [193]. This point represents an important challenge in this analysis, which requires an available 3D prediction model. In addition, for linear B cell epitopes, topology analysis is required in order to localize those located on the protein surface, considering the higher probability of their interaction with the immune system. Lastly, there is a vast number of B cell epitope prediction tools and choosing the best is very hard. It is recommended that a combined method is used for greater precision. We recently published a review about the application of B cell epitope prediction that described a workflow that can be used to identify new targets for the development of fungal infection diagnoses. Additionally, we suggested different methods by which to predict the function and location of proteins [163].

Currently, a robust list of approaches has been proposed in linear and discontinuous B cell epitope prediction [163,194,195]. For linear epitopes, the focus is on the BCPREDS, ABCpred, BepiPred, SVMTriP and CoBepro, which are used in immunogenic and diagnostic studies against *C. albicans* [77], *A. flavus* [186], the *S. schenckii* complex [196], *C. gattii* [197], *Rhizopus oryzae* [198] and *Paracoccidioides* spp. [199,200]. In general, these tools comprise a combination of multiple physical–chemical properties. They are free online via a webserver and are easy to use, requiring the protein sequence in FASTA format.

In the context of the tools used for discontinuous epitopes prediction, there is a group of tools such as BEST, BepiPred-2.0 and CBTOPE that perform the prediction using a primary sequence of proteins. These tools are particularly important when the protein does not have a tertiary structure. On the other hand, tools such as DicoTope [201], SEPPA [202], the BEpro server (formerly known as PEPITO) [203], Ellipro [204] and EPITOPIA [205] use structure-based approaches and require 3D structure information. EPITOPIA yields a higher success rate of 89.4% compared to ElliPro and DiscoTope [205], when used in the analysis of the immunogenic properties of biopharmaceutical enzyme uricase from *A. flavus* and *Bacillus subtilis* [186]. In the other study, SEPPA gave the best performance among the six tools, followed by DiscoTope and BEpro [206].

Like tools for linear epitope prediction, it is difficult to determine the best tool for discontinuous epitope prediction, but it is recommended that tools with a different methodology for analysis are used in order to obtain a better accuracy [207]. It is important to highlight that the prediction of the B cell epitope plays a supporting role alongside the in silico methodology used for vaccine development, while the identification of epitopes for the T cell receptor is the most important goal. In this context, the prediction of the linear B cell epitope can be commonly associated with the initial analysis or associated with discontinuous B cell epitope prediction, which can support the confirmation of the final target once the tertiary structure is elucidated [208,209,210].

### 5.3. Antigenicity Prediction

The use of antigenicity prediction, and its capacity to be recognized by the T cell and B cell receptor, is a particularly important step during the identification of targets for diagnosis and the development of vaccines. Currently, it is possible to cite several tools for this analysis, such as VaxiJen [211], NERVE [212], Vaxign [213], ANTIGENpro [214], the Jenner-Predict server [215], iVAX [216] and VACSol [217]. However, only the VaxiJen v.2.0 [218] was trained with fungal data and because of this, it is used for these pathogens. Its prediction uses the FASTA sequence and is independent of an alignment with experimentally confirmed antigens, but it is based on the physical–chemical properties of proteins with an accuracy of approximately 70% and 89% [211]. This tool was developed in 2007, is available online and nowadays is one of the most cited tools regarding antigenicity, especially in the context of fungal vaccine development (Table 3) [219].

## 6. Concluding Remarks and Perspectives

Fungal infection is a global health problem that is associated with the limitations of therapeutic availability, emergent drug resistance, a large variability in pathogen species, and a difficult diagnosis, which is reflected in a high number of annual deaths.

In view of this, new vaccines and therapeutic approaches represent the best option to circumvent this. Thus, we summarized here the current methodology applied in the development of vaccines against fungal infection. Remarkable progress has been achieved, considering all the major medically important mycoses, by using a variety of vaccine designs in animal studies. A new perspective on the pan-fungal vaccine strategy and on DNA vaccines may represent a promising horizon for the future of fungal vaccine development.

Although great advances have been achieved, there are still several challenges to be overcome. In the current scenario, a safe vaccine for immunocompromised patients (populations most at risk) is required. The diverse sites of infection in the host and the diversity among fungi need to be overcome, and formulations need to be established and commercialization needs to be systematized. Part of this challenge is dependent on science. However, government politics is very important for this progress.

Although it is difficult to determine the tools and methods recommended to better predict vaccine candidates, considering the diversity of fungal infections, we were able to show here a general workflow for in silico analysis. In this sense, we reinforce that these immunoinformatic tools could be the missing piece required in order to successfully identify an effective and safe fungal vaccine candidate.

## Figures and Tables

**Figure 1 jof-09-00633-f001:**
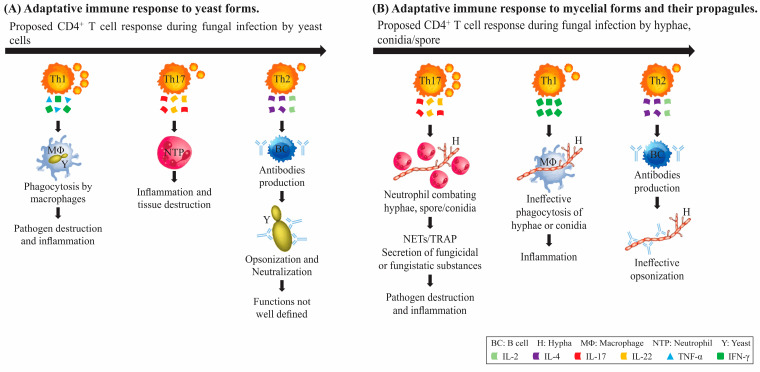
Simplified overview of proposed adaptive immune response to pathogenic fungi. Panel (**A**) illustrates the adaptive immune response to yeast, which necessitates a substantial quantity of the Th1 cell subtype. These cells secrete cytokines, such as IFN-γ, to activate macrophages for phagocytosis, and TNF-α to facilitate granuloma formation, as well as local and systemic inflammatory responses. A regulated response is considered the most effective approach to eliminating pathogenic yeasts. However, the response triggered by the Th17 subtype produces cytokines, such as IL-17, responsible for neutrophil recruitment, and IL-22, which stimulates the recruitment of antigen-presenting cells. During inflammation, the recruitment of neutrophils by Th17 subtypes may cause tissue destruction and aggravate the inflammatory process. Conversely, the response caused by the Th2 subtype results in increased antibody production, which contributes to the opsonization/neutralization of the pathogen. Nevertheless, the efficacy of these functions during pathogenic yeast infections remains undefined. For instance, in individuals with HIV infection, the suppression of CD4^+^ T lymphocytes leads to the host’s inability to eliminate yeast pathogens. Panel (**B**) offers a proposed overview of the adaptive immune response to hyphae, spores, and conidia. In this scenario, the Th17 cell subtype is the most indispensable. As previously mentioned, these cells produce IL-17 and IL-22, which prompt neutrophil recruitment to the inflammation site. Consequently, polymorphonuclear cells secrete various fungicide and fungistatic molecules, including neutrophil extracellular traps (NETs), to eradicate the hyphae. In addition, it triggers inflammatory responses and tissue damage. The Th1 subtype response proves less effective due to the hyphae’s considerable size, rendering phagocytosis by activated macrophages an ineffectual process. Instead, a strong local and systemic inflammatory response ensues. The Th2 subtype response is the least effective, leading to a high production of antibodies. For instance, patients with neutropenia exhibit increased susceptibility to infections caused by fungi in the mycelial form. The arrow depicted in the upper part of the figure represents the frequency of the immune response, with larger arrows signifying a higher occurrence.

**Figure 2 jof-09-00633-f002:**
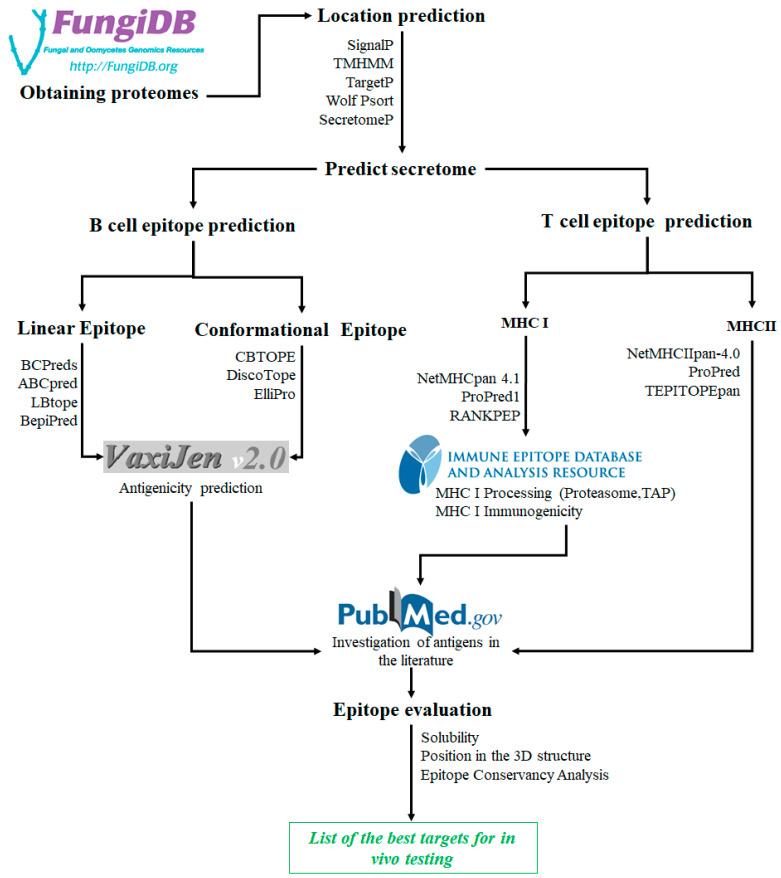
Workflow for prediction of targets for vaccine and diagnosis. Obtention of proteomes by FungiDB or Uniprot; location prediction to find secreted protein; B cell epitope prediction—linear and conformational; antigenicity prediction by VaxiJen; T cell epitope prediction—MHC I (Proteasome, TAP and immunogenicity) and MHCI. Literature investigation: epitope refinement (evaluation)—analyses of solubility, position in the 3D structure and epitope conservancy.

## Data Availability

Not applicable.

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
