# Peer review of "Fungal Vaccine Development: State of the Art and Perspectives Using Immunoinformatics"

_jof, 2023, doi:10.3390/jof9060633_

Round 1

Reviewer 1 Report

Morais Inasio and colleagues described recent advancements in immunoprophylactic fungal vaccines, highlighted the progress of methodological, and experimental immunotherapies against fungal infections. They aimed to describe the use of immunoinformatics to improve challenges in vaccine design. Vaccinology is an important topic; however, we need an updated and thorough review. 

The authors highlight the advancements in silico prediction and immunological evaluation. Morais Inacio and colleagues set forth to organize vast research studies of fungal vaccines. Organizing this work is challenging (ie: type of vaccine, vaccine by the organism, type of immune response required, etc) succinctly without avoiding redundancies.

Unfortunately, readers will have challenges with the organization of this manuscript at this time. It's difficult to follow the structure, the authors need to define sections more clearly, they could benefit from describing the challenges of vaccinations, distinction in the therapeutic treatments and most importantly, the current manuscript is missing many recent publications. Unfortunatley there's redundancies when it's organized by type of vaccine and then described under the pathogen.

Due to critical manuscripts missing and mislabeled, I do not feel that this manuscript is ready at this time. This review may benefit from having a table summarizing the type of vaccine, target host, and if HLA has been considered. 

Major: My main concern is not having the appropriate citations (mislabeled or not referencing the most recent work).

For instance, under inactivated and live attenuated vaccines, although the authors cite Staton and Levitz review (#24), the authors did not add upon more recent work, for instance:(Normile TG, 2022 PMID:35615354)(Fernandes, 2022 DOI: 10.1128/mbio.02328-22)(Lin, 2022 DOI: 10.1128/mbio.02785-21)(Mandel, 2022, DOI:10.1371/journal.ppat.1009832)(Narra, 2016, DOI: 10.1128.IAI.00633-16). Although live/HK vaccines are not the main focus of the manuscript, more recent work should be discussed.

Citing the appropriate manuscripts (ie: 26- incorrect citation for BAD-1: Brandhorst, 1999 PMID:10209038). Citing reviews when describing a single antigen as opposed to the original manuscript may not be appropriate.  

Immunoinformatics was described by Chaudhuri and Ramachandran (DOI: 10.1007/978-1-4939-7104-6_3) and missing in the citation. The authors could describe the improvements using IEDB. 

Under the subunit vaccine listed, some are peptide-based vaccines (ie Hurtgen 2012), while recombinant rCpa1 (Hung 2018, PMID:30104216) is not listed. 

On immunoinformatics, there are key manuscripts describing epitope searches are missing, such as: Specht, 2021,  PMID:35089095- describing MHCII prediction and immunological studies. Dos Santos Dias, 2021, Jensen KK 2018, PMID:29315598  describing NetMHCII, Bhargav, 2017 PMID: 35939284 - computational tools used to identify mucor...

Minor: 

Lines 122-123 highlight safety concerns for the most susceptible invidiuals, it may also be beneficial to address challenges in  vaccine commercialization when using live attenuated or HK vaccines.

Over-redundant citations were observed (ie: citation 29 and 114 are the same), (160 and 206 are the same)(172 and 198 are the same)...

123- it should be H99(gamma) not just H99. 

citation 46 - strain is derived from C. posadasii

Therapeutic studies listed in the title weren't clear in the review, it may be beneficial to separate this or they may be considered to omit it from the title.

I believe IEDB has improved their MHCII prediction model and there's fungal epitopes listed now, this may be highlighted in figure 1.

Author Response

Please find the attached file containing the response letter

Reviewer 2 Report

Dear EIC, Prof. Dr. David S. Perlin

Dear AE, Dr. Estelle Zhang

I hope you are doing well.

This is my review result for manuscript # jof-2215538.

In this study, the authors reviewed probable vaccine approaches against human fungal infections and summarized the methods applied in this era. Although the topic is highly commissioned and there are a lot of published papers, the file should be up-to date. However, I think there is an urgent need for practical works and original papers rather than plenty of repetitive reviews. According to my duty, I carefully read the manuscript several times and listed some minor and major comments below.

Minor comments

  • There are several errors, including language errors, misspellings, improper use of words, and grammatical errors. Carefully review the English and grammar, marking all changes. It is preferable to have a native English speaker do the final editing.
  • The manuscript is supported by 253 references; I think the number of references is very high.
  • Another problem is that only 10 studies, which were published in 2022, and 17 papers, which were published in 2021, are cited in this manuscript (I think the proportion is very low; 10 among 253). So, avoid extra citations by deleting old references and replacing them with newer studies.

Major comments

  • The manuscript should be supported by at least three tables that abstract the data according to the manuscript flow. Tables should be subdivided according to each approach.
  • A simple flow chart should be designed to illustrate the manuscript flow. This can be designed via the Microsoft Power Point tool.
  • Interaction with the host immune response is the most important challenge for the success of a vaccine project. The authors should assay their predicted candidates for different immune scenarios using related tools. Therefore, we need to predict different scenarios for these predicted proteins.
  • The immune pathways should be illustrated with figures. These should be designed by a professional graphic artist using Photoshop or Adobe Illustrator tools.
  • Remove the conclusion section and revise its content under a section named "Concluding remarks and perspectives". This section should discuss the challenges ahead, give suggestions to solve them, and recommend novel approaches.
  • A question that follows my previous comment is: what tools and methods are recommended to better predict promising vaccine candidates?

Author Response

(The authors gave the same response as above.)

Round 2

Reviewer 1 Report

I want to thank the authors for their most recent submission and appreciate their effort to improve the flow of the manuscript while avoiding citation redundancies.

The use of the tables and illustrations to list fungal vaccine and development is a  great idea. I would state that the table is still missing of some more recent work, may I suggest to add the following missing antigens to the Table 1 for Aspergillus and Coccidioides

Aspergillosis: CcpA, live attenuated vaccine https://pubmed.ncbi.nlm.nih.gov/30279286/ 

Antigen Asp f 3 and Asp f 9 (VesiVax Af3/9) vaccine

https://www.sciencedirect.com/science/article/pii/S0264410X22006740?via%3Dihub

optional Aspergillosis: E.coli Nissle 1917 - probiotic bacteria delivered in gastro-resistant hard capsule  that express alpha-galactose protective agains Aspergillosis in pengiuns https://www.frontiersin.org/articles/10.3389/fimmu.2022.897223/full

Coccidioides:

Table 1, reference 20: the correct name for vaccine is known as Formalin Killed Spherules (FKS) 

Δcts2/ard1/cts3 or ΔT - triple attenuated vaccine (citation 21)

GCP-rCpa1 (subunit vaccine) citation 67

Ryp1 - (live attenuated strain) https://pubmed.ncbi.nlm.nih.gov/35385558/

Minor:

Under section 5.1 T cell epitope prediction and 5.2 B cell epitope prediction:

+ The authors may benefit for discussing the challenges between predicting B cell vs T cell epitopes

+ More distinctly, what are the challenges of identifying MHCI vs MHCII epitopes, which illustrates the limited #s of in silico prediction models for MHCII and identifies epitopes deposited in the fungal IEDB site. 

+ The authors could also include the recent World Health Fungal Priority List to the rationale of why fungal vaccine development is needed. https://www.who.int/publications/i/item/9789240060241

line 92: may not need "curiously", this will show up again in line 98 

line 147, although this is an extensive list, I would just state numerous strategies as opposed to counting (40) as not all vaccine strategies have been listed

line 208: italicize Candida

Author Response

Please, find the attached file with the cover letter.

Best regards

Clayton

Reviewer 2 Report

Dear EIC, Prof. Dr. David S. Perlin

Dear AE, Dr. Estelle Zhang

I hope you are doing well.

This is my review result for the revised version of the manuscript # jof-2215538.

  • Kindly revise or re-design Figure 1. This is very poor. A graphical designer should be worked via adobe illustrator or Photoshop tools. Also, we know that TH17 profile is responsible for the inflammatory responses. In figure 1 TH1 profile is presented as responsible for inflammation. Kindly reconsider it.
  • Kindly revise the writing errors; for example, in Figure 1 legend, change formns to forms.
  • As I said in my previous review report, the number of references is very high. Kindly reconsider the old references to remove them from the article.

Author Response

(The authors gave the same response as above.)

Round 3

Reviewer 1 Report

I would like to thank the authors for the latest form the the bioinformatics review as they have included the suggested comments and justified the ones left out.

The review can be published in its most present form. 

Reviewer 2 Report

Dear EIC,

Dear AE,

This manuscript is ready for further consideration for publish processes in your journal.